# A Self-Established “Machining-Measurement-Evaluation” Integrated Platform for Taper Cutting Experiments and Applications

**DOI:** 10.3390/mi12080929

**Published:** 2021-08-04

**Authors:** Xudong Yang, Zexiao Li, Linlin Zhu, Yuchu Dong, Lei Liu, Li Miao, Xiaodong Zhang

**Affiliations:** 1State Key Laboratory of Precision Measuring Technology and Instruments, Laborotary of MicroNano Manufacturing Technology, Tianjin University, Tianjin 300072, China; yangxudong@tju.edu.cn (X.Y.); zexiaoli@tju.edu.cn (Z.L.); l_linzhu@tju.edu.cn (L.Z.); dongyc1996@163.com (Y.D.); 3015202152@tju.edu.cn (L.L.); miaoliii@tju.edu.cn (L.M.); 2Standard Optics Technology Tianjin (SOPT) Company Limited, 80 Fourth Avenue, Binhai New Area, Tianjin 300457, China

**Keywords:** integrated platform, taper cutting, brittle-ductile transition depth, microstructure

## Abstract

Taper-cutting experiments are important means of exploring the nano-cutting mechanisms of hard and brittle materials. Under current cutting conditions, the brittle-ductile transition depth (BDTD) of a material can be obtained through a taper-cutting experiment. However, taper-cutting experiments mostly rely on ultra-precision machining tools, which have a low efficiency and high cost, and it is thus difficult to realize in situ measurements. For taper-cut surfaces, three-dimensional microscopy and two-dimensional image calculation methods are generally used to obtain the BDTDs of materials, which have a great degree of subjectivity, leading to low accuracy. In this paper, an integrated system-processing platform is designed and established in order to realize the processing, measurement, and evaluation of taper-cutting experiments on hard and brittle materials. A spectral confocal sensor is introduced to assist in the assembly and adjustment of the workpiece. This system can directly perform taper-cutting experiments rather than using ultra-precision machining tools, and a small white light interference sensor is integrated for in situ measurement of the three-dimensional topography of the cutting surface. A method for the calculation of BDTD is proposed in order to accurately obtain the BDTDs of materials based on three-dimensional data that are supplemented by two-dimensional images. The results show that the cutting effects of the integrated platform on taper cutting have a strong agreement with the effects of ultra-precision machining tools, thus proving the stability and reliability of the integrated platform. The two-dimensional image measurement results show that the proposed measurement method is accurate and feasible. Finally, microstructure arrays were fabricated on the integrated platform as a typical case of a high-precision application.

## 1. Introduction

With the development of technological applications, the precision of various optical lenses, prisms, and other optical devices is required to reach the nanometer level [1]. At present, ultra-precision processing is mainly used for processing and preparation [2]. In the processing of various hard and brittle materials, due to their natural characteristics, there is still a range of problems, such as easy cleavage, brittleness, and anisotropy in the performance of ultra-precision processing [3,4,5]. However, when the cutting depth of a material is small enough (usually at the sub-micron level), brittle materials will undergo a change from brittleness to plasticity. This phenomenon is of great significance for the realization of ultra-smooth surfaces of brittle materials [6,7,8]. In order to determine the critical cutting depth for the removal of the brittleness and plasticity of a material, a taper-cutting experiment is the main method for effectively assessing the actual cutting thickness of the material—from shallow to deep. It is widely used in the study and realization of the cutting mechanisms of hard and brittle materials, and it is an important part of ultra-precision machining [9,10].

The commonly used nano-taper-cutting experiments are mainly carried out with ultra-precision machining tools. The linear motion accuracy can reach tens of nanometers. When the parameters of the cutting process are reasonably set, a diamond tool can follow a set path to realize the taper angle with respect to the material’s surface in the process of removing the processed material in a downward direction. This method can be used to obtain the cutting thickness (from nanometers to micrometers) so as to analyze the the mechanisms of deformation and removal of the material with different cutting thicknesses [11,12] simultaneously with the BDTD of the material under the current cutting conditions. For example, Fang et al. used ultra-precision machining tools to conduct taper-cutting experiments on single-crystal silicon, single-crystal germanium, etc., to analyze the cutting characteristics of the materials [13,14]. Xiao et al. used ultra-precision machining tools to conduct taper-cutting experiments on 6H-SiC and presented an investigation of the mechanism of the brittle-ductile cutting mode transition [15]. After a taper-cutting experiment is completed, it is necessary to characterize the tapered surface of the material in order to calculate the BDTD under the current cutting conditions. Currently, two-dimensional image calculation methods or three-dimensional microscopic observation methods are generally used for characterization [16,17,18]. Two-dimensional image processing calculation methods use the geometric relationship between the tool and the chute, judge the position of the brittle-ductile transition according to the changes in the material’s surface, and then calculate the BDTD. Wen Gu et al. improved the assessment of the brittle-ductile transition based on the surface conditions using a method of changing the position to observe the two-dimensional plane with an optical microscope and then calculating the BDTD of the material by extracting the envelope curve [19]. Three-dimensional microscopic measurement methods mainly use a scanning electron microscope (SEM), atomic force microscope (AFM), transmission electron microscope (TEM), etc., as well as a white light interferometer, laser confocal measuring instrument, or other three-dimensional scanning equipment for performing measurements; then, according to the three-dimensional information, the position of the brittle-ductile transition point is judged, and the BDTD is obtained [20,21,22,23]. For example, Fang et al. used an SEM and AFM to observe a surface after taper cutting and obtained the critical cutting thickness of the brittle-ductile transition during the nano-cutting of single-crystal silicon [24,25].

Although taper-cutting experiments have been conducted with the methods described above and observations of the grooves have been made, there are still several problems with these methods, stemming from the fact that cutting experiments are conducted with ultra-precision machining tools. When the cutting experiments are conducted with ultra-precision machine tools, the conditions of which are harsh and the cost of which is high. Moreover, it is difficult to integrate sensors to realize on-line rapid measurement. When the processed surface need to be observed, measured and evaluated after the taper cut, the measuring equipment used is usually large in size. You have to remove the workpiece and observe it at the offline measuring equipment, which is time-consuming and laborious, and the accuracy of current methods is not accurate enough to reflect the nano-cutting characteristics of the material well. Currently, there is a significant need for methods and equipment that can reduce the costs of taper-cutting experiments, achieve high-precision online observations and measurements, and improve the efficiency of such experiments.

In this paper, a linear motor displacement table and integrated measurement sensor were used, and a “machining-measurement-evaluation” integrated system platform was designed and constructed. The platform has low cost and high flexibility, which is applied on an efficient experimental method for taper-cutting instead of ultra-precision machine. At the same time, the system platform is used to perform a three-dimensional measurement on the surface of the tapered surface, and a measurement evaluation algorithm process is proposed to quickly and efficiently obtain the BDTD of the material. Taper-cutting experiments with the same parameters were carried out on single-crystal germanium materials using ultra-precision machining tools and the system established here. The results of the comparisons are basically consistent, which proves that the system is stable and the method is feasible and effective.

## 2. Taper-Cutting Experiment and Measurement Evaluation Method

The research route is shown in Figure 1. By analyzing the requirements of the taper-cutting experiment, an integrated platform of “processing-measurement-evaluation” is designed. In order to realize the adjustability of the surface slope of the workpiece, the system incorporated a spectral confocal displacement sensor that was used for the measurement of the inclination angle of the workpiece’s surface, as well as the use of a fixed tool to cut the workpiece (with cuts ranging from shallow to deep) in order to complete the taper-cutting experiments. For the high-precision three-dimensional measurement of the processed surface, a small white light measurement sensor was integrated in order to perform high-precision measurements, and the measurement results were processed to provide data. An algorithm process based on three-dimensional data and two-dimensional images is proposed to extract the BDTD of materials with high precision under the current cutting conditions. In addition, the system platform can also manufacture microstructure arrays.

### 2.1. Method of the Taper-Cutting Experiments

This paper proposes the use of an integrated spectral confocal sensor as part of the platform in order to measure the feedback adjustment of the angular displacement stage, change the fixed inclination angle of the workpiece’s surface, and then control the workpiece to perform a single-axis linear motion relative to the diamond tool in order to realize the removal of material by the tool and complete the taper-cutting experiment.

As shown in Figure 2, the specific implementation method was as follows: Under the condition that the surface of the workpiece was relatively flat, data collection was performed by a high-precision displacement sensor that was fixed above the workpiece. In order to avoid random errors of individual coarse points in the measurement results, the current partial square area was selected as the measurement range, the sets of the height values of the workpiece and the sensor surface were recorded with a fixed scanning track, and the average value of the height value $h_{1}$ in the measurement area was calculated according to the height value of the center point. Then, the workpiece was moved in the horizontal direction for a distance *d*. The height value $h_{2}$ of a square area of the same size at the other end of the workpiece was measured in the same way, the distance between the center point of the area and the surface of the workpiece was obtained, and the inclination angle θ of the workpiece was calculated according to
(1)$$\theta =\arctan\left(\frac{h_{2}-h_{1}}{d}\right)$$

Depending on the calculated inclination angle of the workpiece, the angular displacement stage was adjusted; then, the angle was measured with the displacement sensor, and this process was repeated until the inclination angle of the workpiece met the requirements of the taper-cutting experiment. With this method, the taper-cutting experiment only required that the knife had a degree of freedom of movement in one direction. At the same time, this method could be realized by using a general-precision guide rail and did not need to rely on ultra-precision machining tools, which greatly reduced the cost and complexity of the experiments.

It should be noted that the measurement accuracy of the tilt angle of the workpiece was related to the horizontal movement distance and the resolution of the displacement sensor. By increasing the distance between the two measurement positions, the measurement accuracy of the tilt angle could be significantly improved. For example, the surface slope of the workpiece was 1:1000, and the angle was 0.057${}^{\circ }$. When measuring the slope of the workpiece, if the workpiece was moved horizontally by 1 mm, its vertical height would change to 1 μm. Under the condition that the workpiece’s surface was guaranteed to be flat, if the workpiece was moved horizontally by 10 mm, its vertical height would change by 10 μm. When calculating the angle measurement resolution based on the same sensor resolution, the measurement resolution was significantly improved, and the accuracy of the measurement of the inclination angle of the workpiece could be improved. In addition, in order to improve the accuracy of the calculation of the angle, multiple repeated measurements could be performed to eliminate the influence of accidental errors.

The above is a single-direction angle adjustment method. For the adjustment of the plane angle, any deviation of the inclination angle can be decomposed into the deviations in the two directions of the X and Y axes, so two perpendicularly connected angle translation stages can be used for adjustment. The angles of the two angular displacement stages need to be adjusted in the X and Y-axis directions, and finally, the adjustment of the tilt angle of the workpiece in the two directions can be realized.

### 2.2. Measurement and Evaluation Method for the BDTD

In this paper, a self-developed miniaturized white light interferometric sensor is used; it is small and can be integrated in situ. When using this sensor to measure the three-dimensional topography of a cut surface, the microscopic three-dimensional measurement accuracy is higher, and the research on the nano-cutting mechanisms of materials can be of greater significance. Through the algorithmic processing of the three-dimensional surface topography, the brittle-ductile transition position of the material is judged, and then the brittle-ductile transition depth is obtained. This method can be used to accurately locate the brittle-ductile transition point and to realize the automatic and high-precision calculation of the brittle-ductile transition depth of a chamfered surface.

White light interference is a very common method for the observation and measurement of microscopic topography, but its angular characteristics are poor and the numerical aperture value is small, which is not suitable for the measurement of rough surfaces. Due to the taper-cutting experiment, in the brittle removal area, the fragmentation of the material is serious, and the surface is rough. If the measurement method of white light interference is used, the information will be distorted if this part of the information is not processed, as shown in Figure 3. It can be seen that there is obvious noise and distortion on the surface, and data processing is thus required.

The white light interferometry data obtained must be leveled first. The leveling process uses the Levenberg–Marquardt least squares optimization algorithm to fit the plane of the data. After the fitted plane is obtained, the plane is subtracted from the data to complete the calibration of the leveling of the measurement data.

For the removal of gross error points, the geometric characteristics of most points can be represented by their domain points. White light interferometry data are typical grid point cloud data. Therefore, statistical methods can be used to quickly eliminate gross error points in this research.

Suppose that a point in the white light interferometry data is $p_{i}$, and the set of points in its four neighborhoods is $N\left(p_{i}\right)=\left\{p_{i1},p_{i2},p_{i3},p_{i4}\right\}$. In the coarse error point elimination algorithm, for each point $p_{i}$, the average value di of the spatial geometric distances of all points in its four neighborhoods $N(p_{i})$ is calculated. Then, the average distance μ and standard deviation σ of all measured data points $d_{i}$ are calculated, and the points whose average distances are outside the threshold range t are regarded as gross error points and eliminated. The specific flow of the algorithm is as follows:(1)Considering the characteristics of the collected 3D point cloud and the calculation efficiency of the algorithm, for the data point $p_{i}$, calculate the average distance $d_{i}$ between it and its four neighbouring points, which is expressed as
(2)$$d_{i}=\frac{1}{4}\sum_{j=1}^{4}∥p_{i}-p_{ij}∥$$(2)Calculate the mean μ and standard deviation σ of the average distance $d_{i}$ of all data points $p_{i}$.(3)Judge the size relationship between the average distance $d_{i}$ of $p_{i}$. If it satisfies
(3)$$\mu -t\sigma \leq d_{i}\leq \mu +t\sigma$$
the point is considered to be a normal data point and retained. If the following formula is not satisfied, it means that the numerical deviation of this point is large, and it is a gross error deletion. Considering the characteristics of the white light interferometric sensor and the conclusions of different *t* value experiments, the value of the threshold *t* is generally set to 2 in this algorithm.

The data after removing the gross error points must be processed through median filtering to smooth the surface data in order to facilitate subsequent analysis and processing. In data processing, if the data are strictly processed according to the definition of median filtering, the median value of the data points within each of their filtering ranges must be calculated. This process is very computationally intensive when the filtering range is large. The efficiency of the processing of the measurement data is low. Therefore, in order to ensure the processing speed of the median filter, the Finite impulse response Median Hybrid (FMH) filter is used in this research.

In the FMH filter, the row data or column data of the measurement data are processed with the $H_{M}\left(z\right)$ filter to obtain an approximate median filtering result. An FMH filter with a length of M=3 is used in this research. This filter approximately converts the median filter for the mean filter calculation so that the median filter processing of the data can be completed through the convolution calculation, which greatly speeds up the data processing speed. For two-dimensional white light interferometric data, the FMH filter first processes each row of the data, and then calculates each column of the data to complete the median filtering of the two-dimensional data. The FMH filtering process is as follows:(1)Perform average filtering on row or column data with three sub-filters. Let *k* be the median filtering range, and the three sub-filters are
(4)$$H_{1}\left(z\right)=\frac{1}{k}\left(z^{k}+z^{k-1}+\cdots +z^{1}\right)$$
(5)$$H_{2}\left(z\right)=1$$
(6)$$H_{3}\left(z\right)=\frac{1}{k}\left(z^{-1}+z^{-2}+\cdots +z^{-k}\right)$$(2)The median value of the filtering results of the three sub-filters of each data point is calculated as the median filtering calculation result for that point.

The overall process and results of the data processing are shown in Figure 4. The chute depth corresponding to the brittle-ductile transition position is the BDTD of the material under the current processing conditions. In order to obtain the BDTD of the material under the current cutting conditions, it is first necessary to determine the position of the brittle-ductile transition point on the surface of the material after taper-cutting. Based on the three-dimensional point cloud from the white light measurement and the original gray-scale image taken by the camera after the processing with the noise reduction filter, a series of supplementary image processing steps and three-dimensional feature extraction (as the main focus of the extraction of the depth of the bevel and brittle-ductile transformation) are proposed.

The process of the algorithm is shown in Figure 5. According to the image used and the white light three-dimensional solution, graphics processing, such as boundary extraction, is performed to roughly locate the area where the brittle-ductile transition point is located, and then the contour extraction and fitting are performed in the corresponding area of the three-dimensional point cloud. During the process, precise positioning is carried out with the help of features such as the deep mutation of the brittle-ductile transition point. After determining the position of the brittle-ductile transition point, by comparing the height difference between it and the fitting plane, the brittle-ductile transition depth of the material under the current cutting conditions is calculated.

The image-assisted solution process is shown in Figure 6. The original image has a certain amount of ambient light interference, which does not affect the three-dimensional shape of the white light solution, but it will cause greater interference in the boundary extraction. Therefore, first, sliding median filtering and gray-scale transformation are performed on this image, in turn, in order to obtain the ambient light information matrix at each pixel of the image; then, the ambient light matrix is subtracted from the original image to obtain the result of the ambient light homogenization, as shown in the figure. After processing, the edge information of the processed groove is more obvious. Then, the Laplace transform is used to extract the upper and lower edges of the chamfered groove, as well as to fit the center line and calculate the inclination angle of the chamfered groove θ. The gradient operator can characterize the dramatic changes in the gray values of image pixels well, but it has a poor effect when the gray value changes slowly; the Laplacian operator is the second-order partial derivative of the image in the X and Y directions. Their sum has good versatility, and it is calculated as
(7)$$\Delta f=\frac{\partial^{2}f}{\partial x^{2}}+\frac{\partial^{2}f}{\partial y^{2}}$$

In order to correctly extract the area where the brittle-ductile transition point is located, the image is rotated accordingly so that the cutting groove is basically parallel to the X-axis. The rotated image and the corresponding edge extraction are shown in Figure 7a. It is observed that when the chamfered chute is extracted from the edge, there will be an obvious edge at the brittle-ductile transition position on the inner wall of the chute, and the area where the transition point is located—that is, the range of the red frame—can be determined, as shown in Figure 7b.

Because the three-dimensional point cloud has a one-to-one correspondence with the image pixels, the processed groove point cloud can also be turned parallel to the X-axis according to the rotation angle calculated above, as shown in Figure 7c. The area in the red frame corresponds to the area in Figure 7d, which is the area where the brittle-ductile transition point is located. A leveling process is performed on the rotated three-dimensional point cloud to make the plane area parallel to the XOY plane. At this time, the height difference between the brittle-ductile transition point and the plane along the Z direction is the BDTD of the material when subjected to this processing technology.

According to the shape of the diamond tool and the chamfering mechanism, the bottom of the cutting groove is elliptical. In order to accurately locate the groove contour of each section line in the YOZ plane, the contour of the groove section is reconstructed with a circle-fitting method. As shown in Figure 8a, in the YOZ plane, because the cutting depth of each section is different, the number of points used for fitting will directly determine the accuracy of the fitting. Therefore, an algorithm with a dynamic sampling length is adopted to dynamically select the interval for fitting according to the cross-sectional profile, thereby achieving the purpose of better approximating the real profile, as shown in Figure 8b.

After fitting the profile of each section and extracting the lowest point, the projection of the lowest point profile on the XOZ plane of the groove in the red area in Figure 7d is shown in Figure 8c. The material is at the brittle-ductile transition point. The height will drop significantly, and the brittle-ductile transition point position is determined based on this feature; then, the BDTD is calculated.

## 3. System Design and Construction

According to the processing requirements of the taper-cutting experiment, it was determined that the rigidity and modal performance of the processing system met the requirements of the taper-cutting experiment. The processing and measurement modules were integrated, and the integrated micro-structural platform for cutting and measurement was established through engineering. The physical object is shown in Figure 9. The parameters of the system are listed in Table 1. The overall structure adopted a marble gantry bracket as support, which had a large weight, low thermal deformation, and good rigidity and stability; the moving part of the experimental platform consisted of three orthogonal air-floating platforms (*x*-axis, *y*-axis, and *z*-axis) and a high-precision electric turntable that was composed of an X/Y-axis orthogonally placed above the marble platform and a *z*-axis that was vertically fixed on the horizontal frame of the marble gantry. All stages used a grating ruler as a reference for the displacement distance feedback in order to ensure the positioning accuracy of the platform; the three-axis XYZ stages were driven by linear servo motors with a high-frequency response, high motion positioning accuracy, smooth motion process, and no generation of severe chattering; the tool was a diamond tool that was fixed on the *z*-axis with a tool holder, and the measurement modules of the spectral confocal (SC) sensor (Thinkfocus company, Shanghai) and a self-made in situ white light interference (WLI) sensor were also installed on the *z*-axis. The movement axis of the system platform drove it to move. For the measurement of the workpiece, one or two orthogonal angular displacement stages were fixed on the X/Y table, and the workpiece to be tested was fixed on it in order to realize the multi-directional angle adjustment of the workpiece. Four air cushions were fixed under the overall workbench to block environmental vibrations and thus effectively avoided the impact of the environment on vibrations during processing. When the platform performed cutting operations, the workpiece on the worktable moved linearly along the *x*/*y*-axis, the diamond tool on the tool holder moved linearly along the *z*-axis, and the platform controlled the tool and workpiece in their movement along a fixed path through the contact of the tool with the workpiece and the generation of relative movement; thus, the process of the taper-cutting experiment was completed, and the results of the cutting process could finally be obtained.

Figure 10 presents a diagram of the hardware structure of the control system. The overall hardware is divided into two parts: the processing module and the measurement module. The two modules share the same set of motion axis systems. The hardware control mainly uses the communication between the upper computer and the lower computer to realize the control. The main body of the lower computer is a three-part controller, including a multi-axis motion controller, a white light interferometric sensor controller, and a spectral confocal controller. As shown in the red dotted frame, the system’s motion axis is the core of the entire system, and its motion positioning accuracy ensures the processing and measurement accuracy of the entire system. The motion axis system is controlled by the multi-axis system power controller. Due to its powerful data processing capabilities, it is mainly responsible for real-time tasks, such as motion control, interpolation operation, program execution, and status monitoring of each axis. The multi-axis motion controller controls the linear stages of the X-axis, Y-axis, and Z-axis, as well as the rotary axis stages through channel 1, channel 2, channel 3, and channel 4, respectively; it also realizes the real-time positioning of the axis system with the grating ruler and limit sensor, as well as compensation and limitation of the stroke of the movement. The tool is fixed on the Z-axis, the angular displacement stage is fixed on the XY-axis, and the fixed workpiece and the tool move relative to each other to realize the cutting of the workpiece. The measurement module includes a spectral confocal sensor and an in situ white light interference sensor. Both adopt a miniaturized design and a small size to achieve system integration. They are installed on the Z-axis like the tool; they are used to adjust the posture of the workpiece and the tool, as well as to adjust the tool. The measurement and evaluation of the results take place after processing.

## 4. Experimental Process and Results

As shown in Figure 11, the X/Y table depicts a hard and brittle workpiece that will be processed with a flat surface. The workpiece is measured with a spectral confocal sensor. The tilt table is adjusted to make the slope of the workpiece surface in the processing direction meet the design requirements, and the workpiece is moved to the bottom of the tool. The tool is then moved horizontally, and the tool is lowered onto the Z-axis for the taper-cutting experiments.

After the workpiece is installed, the surface slope of the workpiece is adjusted according to the method mentioned above so that the surface slope of the workpiece reaches the preset angle. Then the tool needs to be set so that the tool is infinitely close to the surface of the workpiece and so that the tool moves along the designed path. It can be seen with an optical microscope if the tool is in contact with the surface of the workpiece at the bottom of the movement. If there is no contact, the tool will drop a certain distance (generally set to 0.5–1 μm) and continue to move along this path until the optical microscope observes obvious grooves on the surface of the workpiece. It is thus proved that the tool is in contact with the surface of the workpiece, and the taper-cutting experiment is completed.

### 4.1. Taper-Cutting Experiment and Comparison of Results

In order to prove the stability of the method and the experimental platform, an ultra-precision machining tool and the experimental system were used to conduct taper-cutting experiments on the same material with the same parameters to observe whether there were obvious differences in the morphological characteristics of the experimental processing grooves; then, the BDTDs of the materials with these processing parameters were calculated and compared. The experimental parameters are shown in Table 2.

The comparative experimental results are shown in Figure 12, where Figure 12a,c,e show 2D gray-scale images, 3D measurement topography, and 2D profile of the taper-cutting experiments conducted by the self-established system, respectively. In the meantime, Figure 12b,d,f present the taper-cutting experiments conducted by ultra-precision machine tools. It can be seen that the results of the taper-cutting experiments of the two systems were basically the same, showing a groove that was obviously removed obliquely downward, and the brittle-ductile transition area was clear and obvious. The results of the three-dimensional measurements of the chutes of the two systems were obtained using the small white light interferometric sensor integrated into the system. According to the calculation method mentioned above, the two-dimensional cross-sectional profile of the chute was obtained, and the calculated BDTD was approximately the same as that of germanium. The theoretical brittle-ductile value was close to the BDTD, which was about 90 nm. The above proves that the system is stable and reliable and that the measurement method is feasible and accurate.

### 4.2. Application—Microprism Array Preparation

The integrated cutting and measurement platform designed and established in this article could perform not only taper-cutting experiments on the material but also microstructure processing. Figure 13a presents a diagram of the system for processing the microprism. The tool was replaced with a sharp knife with a specific angle, a rotating platform was fixed under the workpiece, and the material was removed by planing. The surface of the workpiece to be processed was measured with the confocal sensor for auxiliary leveling. After the cutting tool was completed, the material was removed in one direction. Then, the workpiece was rotated, and the process was repeated with two rotations of 60${}^{\circ }$. The final processing results are shown in Figure 13b. The microstructure was uniform in size, the surface was smooth, and the surface had a high shape accuracy; thus, it could be used as a mold for the preparation of reflective films.

## 5. Conclusions

Currently taper-cutting experiments on hard and brittle materials are mostly based on ultra-precision machining tools, which have a high cost and low efficiency. It is difficult to realize online in situ measurements, and the accuracy of the calculation of the BDTD is low, which is not conducive to research on nano-cutting mechanisms. This paper describes the design and construction of an integrated cutting and measurement platform that uses white light interference to measure and calculate the BDTD; this study also compared the taper-cutting results of ultra-precision machining tools and the self-established system under the same cutting parameters, thus proving that the system is stable and suggesting that this measurement evaluation method is feasible. The main conclusions are summarized as follows:(1)An integrated cutting and measurement platform was designed by using an angular displacement stage to adjust the posture of the workpiece, using a spectral confocal sensor to measure the slope of the workpiece and adjusting it in real time so as to realize the tool’s oblique downward removal of the material and obtain high-quality concavity in the slot used for observation and measurement.(2)A small white light sensor was used for in situ observation and integrated into the integrated cutting and measurement platform to directly obtain the three-dimensional shape information of the material, thus aiming at the problem of brittleness by removing the regional distortion, optimizing the white light solution, and improving the display effect. The surface was leveled, and the BDTD of the material could be quickly and accurately obtained.(3)An experimental platform was built, and the results were compared with those of an ultra-precision machining tool to prove that the platform was stable and reliable and that the proposed measurement evaluation method was feasible. At the same time, the platform was also shown to be able to perform high-precision preparation of microstructure arrays.

## Figures and Tables

**Figure 1 micromachines-12-00929-f001:**
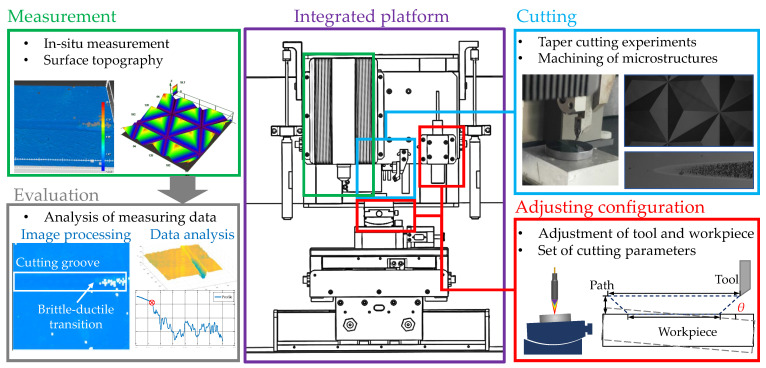
The research route.

**Figure 2 micromachines-12-00929-f002:**
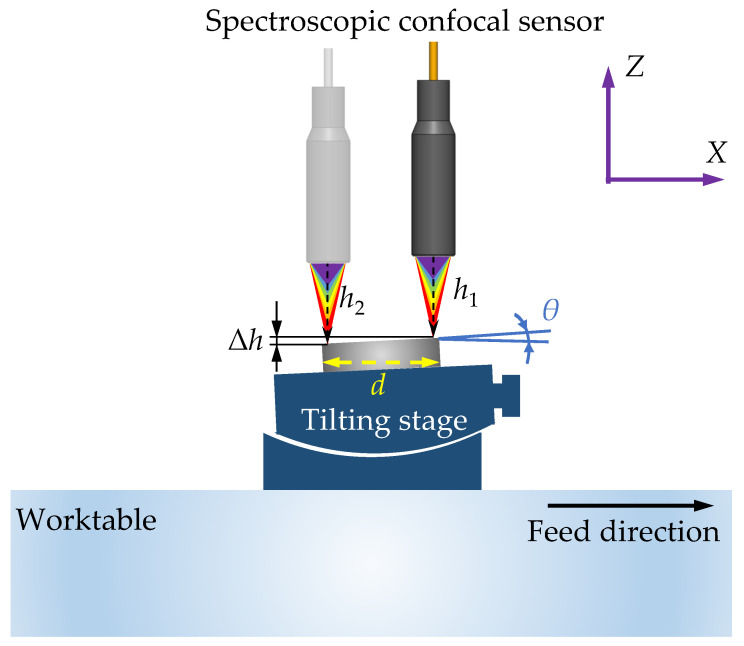
The method of the taper-cutting experiments.

**Figure 3 micromachines-12-00929-f003:**
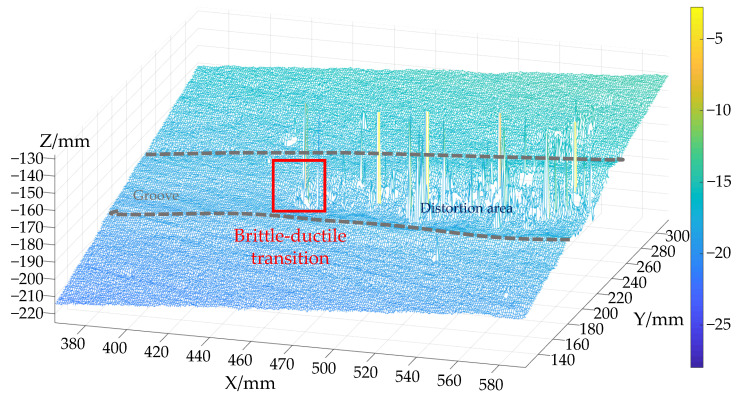
Groove distortion with white light interferometry.

**Figure 4 micromachines-12-00929-f004:**
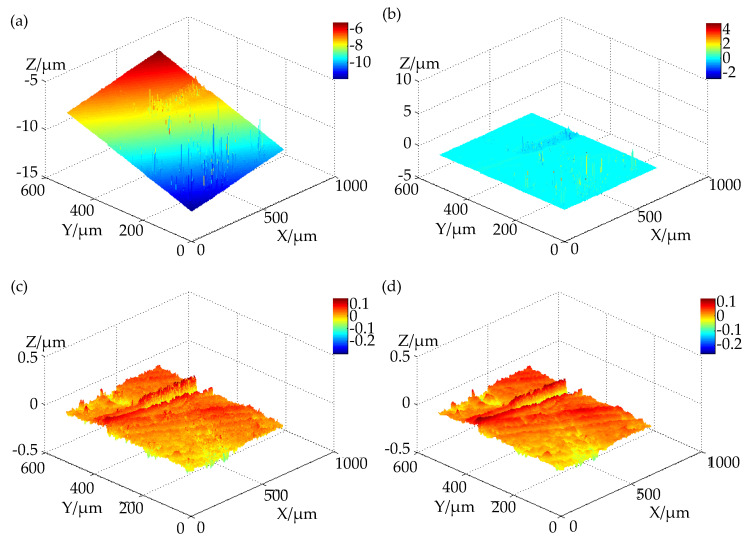
Process and results of data processing: (**a**) original data; (**b**) leveling; (**c**) removing gross errors; (**d**) median filtering.

**Figure 5 micromachines-12-00929-f005:**
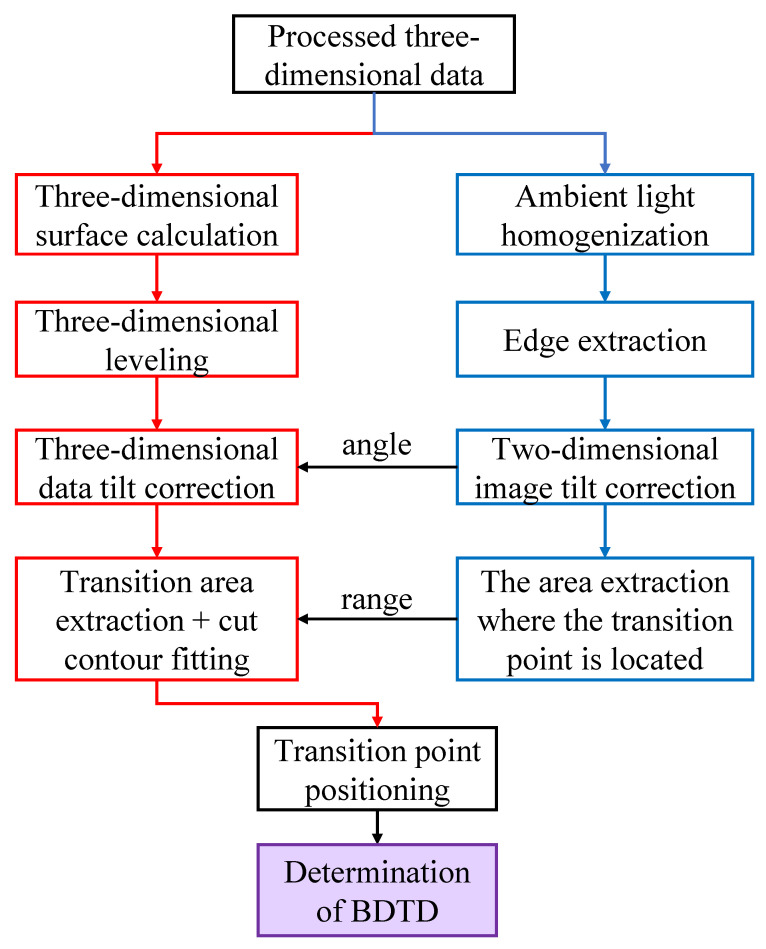
The process of the algorithm for the positioning of the brittle-ductile transition point.

**Figure 6 micromachines-12-00929-f006:**
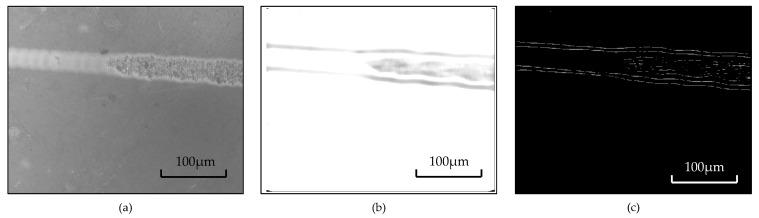
The image processing process of the taper-cutting result: (**a**) original image; (**b**) ambient light homogenization result; (**c**) Laplacian edge extraction results.

**Figure 7 micromachines-12-00929-f007:**
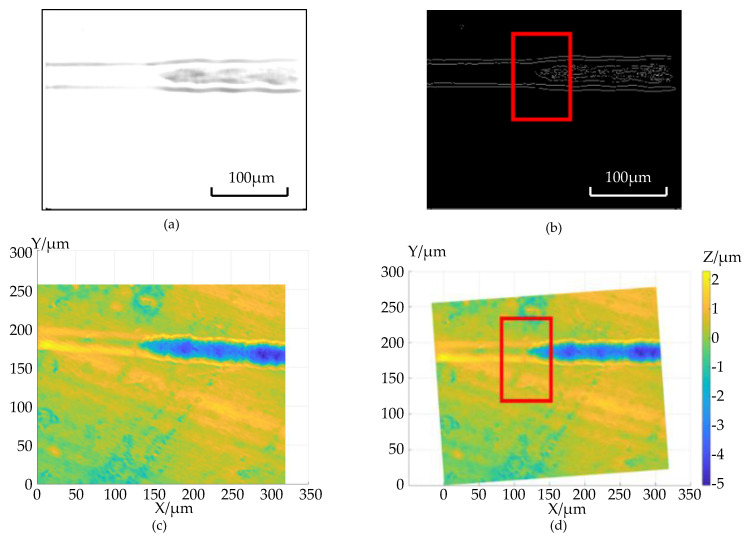
The process of determining the position of the brittle-ductile transition: (**a**) 2D image after rotation. (**b**) Laplacian edge extraction results. (**c**) 3D point cloud data. (**d**) 3D point cloud data after rotation.

**Figure 8 micromachines-12-00929-f008:**
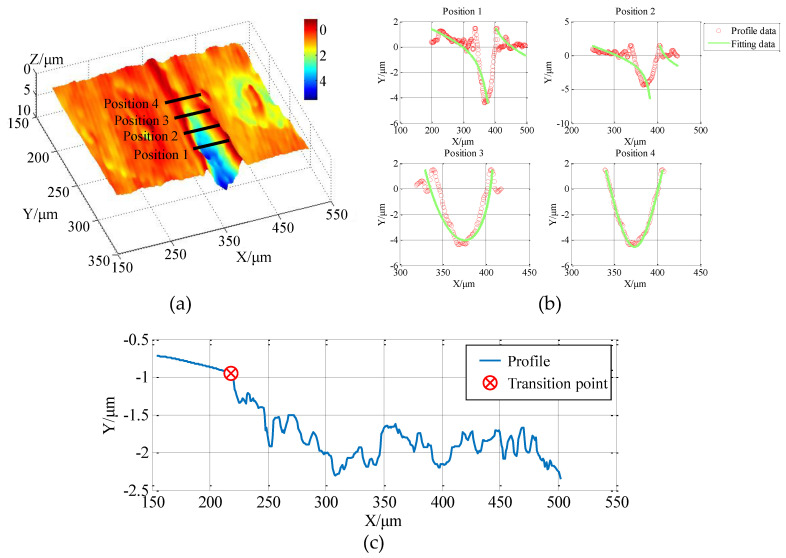
The process of determining the BDTD: (**a**) 3D point cloud of the area of the brittle-ductile transition point; (**b**) the sectional profile at different positions; (**c**) a projection of the lowest point of the cut contour.

**Figure 9 micromachines-12-00929-f009:**
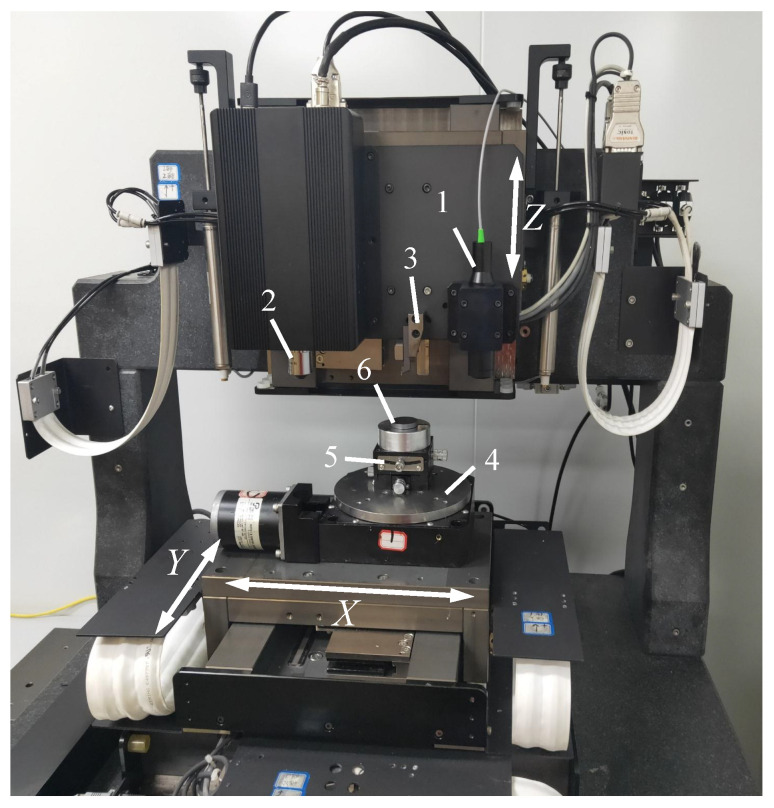
Integrated Platform. 1: Spectroscopic confocal sensor; 2: White light interference (WLI) sensor: 3: cutting tool; 4: rotating stage; 5: tilting stage; 6: workpiece.

**Figure 10 micromachines-12-00929-f010:**
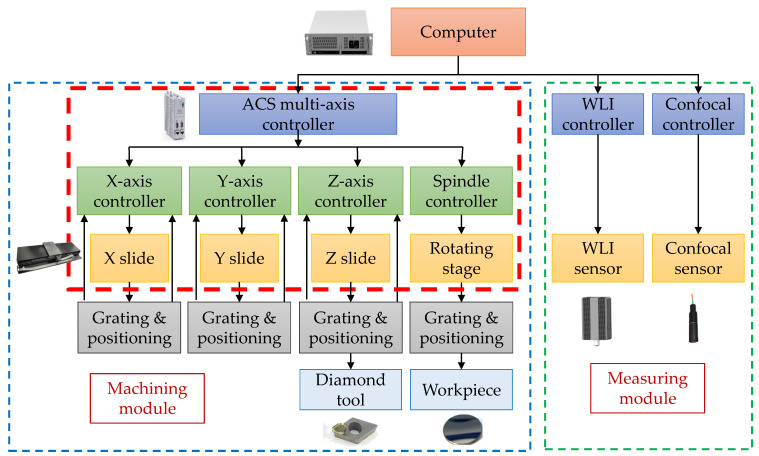
System control block diagram.

**Figure 11 micromachines-12-00929-f011:**
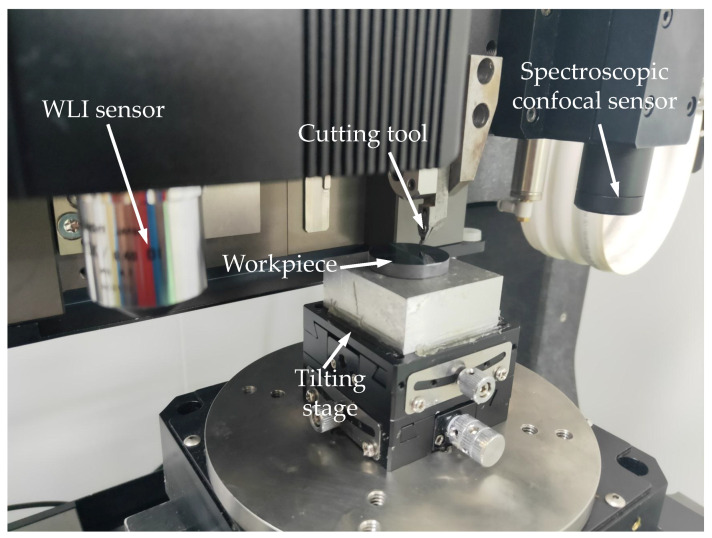
The scene of a taper-cutting experiment.

**Figure 12 micromachines-12-00929-f012:**
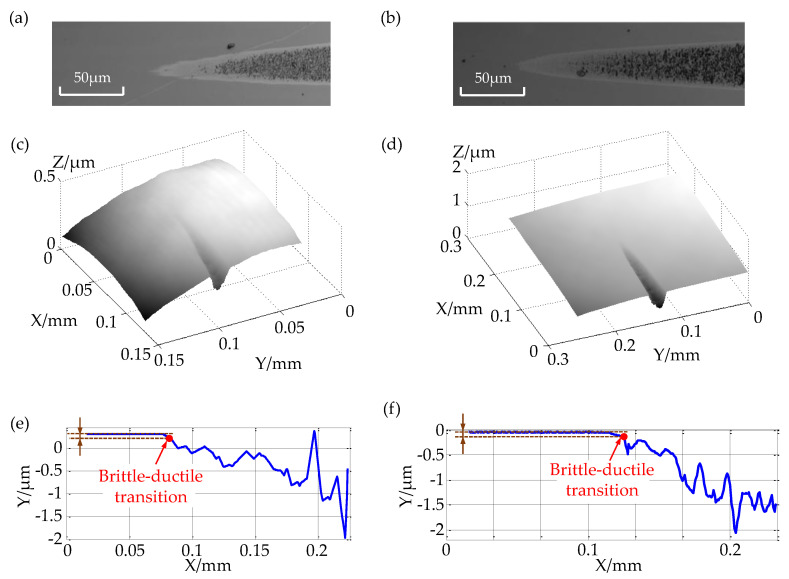
The results of the taper-cutting comparison of (**a**) 2D gray-scale images of taper cutting with the integrated cutting and measurement platform; (**b**) 2D grayscale images of taper cutting with the ultra-precision machining tool; (**c**) 3D measurement results of taper cutting with the integrated cutting and measurement platform; (**d**) 3D measurement results of taper cutting with the ultra-precision machining tool; (**e**) 2D profile of the taper cutting with the integrated cutting and measurement platform; (**f**) 2D profile of the taper cutting with the ultra-precision machining tool.

**Figure 13 micromachines-12-00929-f013:**
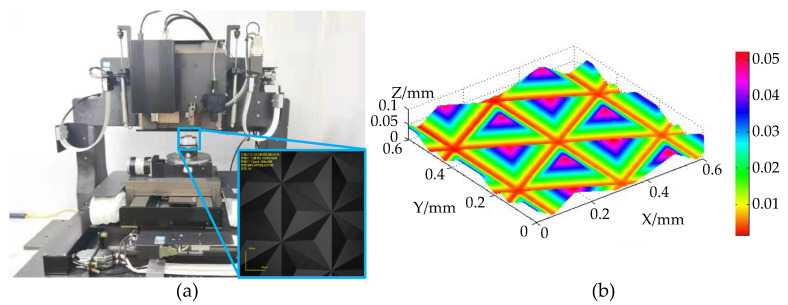
Microprism manufacturing. (**a**) The scene of the manufacturing of a microprism array. (**b**) The results of the 3D measurement of the microprism array.

**Table 1 micromachines-12-00929-t001:** The hardware parameters of the system.

Hardware	Travel and Range	Accuracy	Others
X/Y/Z axes	200 mm	200 nm	/
SC sensor	55 μm	±0.02% Full-scale	Numerical aperture 45${}^{\circ }$
WLI sensor	270 μm × 337 μm	10 nm	Numerical aperture 40${}^{\circ }$

**Table 2 micromachines-12-00929-t002:** The experimental parameters for the comparison.

Parameters	Value
Material	Monocrystalline germanium
Tool noise radius	1.507 mm
Tool rake angle	0${}^{\circ }$
Cutting speed	2 mm/s
Tilt angle	≈1:3000
Maximum cutting depth	1–2 μm

## Data Availability

The data are not publicly available because the data also forms part of an ongoing study.

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
