# Peer review of "A Self-Established “Machining-Measurement-Evaluation” Integrated Platform for Taper Cutting Experiments and Applications"

_micromachines, 2021, doi:10.3390/mi12080929_

Round 1
Reviewer 1 Report
The paper makes important contributions in terms of measuring the depth of brittle-ductile transition (BDTD) of a material. Starting from the disadvantages of the classical method, based on conical cutting experiments, the authors propose an integrated cutting and measuring platform. This platform measures and calculates the brittle-plastic transition depth, using the white light interface. The authors compare the results obtained by the two methods and find that they are very close, proving that the proposed method is feasible.
The authors may make some corrections.
a) The authors could improve Figure 2 by representing the h1 and h2. It could also specify in the text what these represent;
b) The authors could explain (on page 5) why they chose 4 measurement points (relation (2))
c) The authors could use only one of the relations (3) and (4) which are identical. This relationship should also be referred to in the text, including by justifying the value 2 for parameter t.
d) The authors could be clearer, in lines 138-141, by explaining whether it is a matter of moving the piece or the material.
e) on page 13, row 352, reference is made to Figure 12 not to figure 11
f) the authors could give more precise indications on the characteristics of the component elements of the proposed platform (manufacturer, precision, etc.).
Reviewer 2 Report
- I'm completely missing discussion in this manuscript.
- Author should clearly describe, what kind of novelty brings his BDTD system in compare with others. Because I do not think, that system in manuscript solved all mentioned in the end of Introduction: "The experimental conditions are harsh and the costs are high. There are many limitations in the actual experimental process. When a processed surface is observed, measured, and evaluated after a taper cut has been completed, the measuring equipment used is usually large in size. One must remove the workpiece and observe it with the offline measuring equipment. If it is necessary to continue the experiment, it must be installed and adjusted again. Repeating this process is time-consuming and laborious, and the accuracy is not high enough to reflect the characteristics of the nano-cutting of the material well."
- Some images does not have scale.
Round 2
Reviewer 2 Report
Author reflected all my comments and improved his manuscript.